# Inhibition of miR-96-5p in the mouse brain increases glutathione levels by altering NOVA1 expression

Chisato Kinoshita [1], Kazue Kikuchi-Utsumi[1], Koji Aoyama[1], Ryo Suzuki[2], Yayoi Okamoto[1,3], Nobuko Matsumura[1], Daiki Omata[2], Kazuo Maruyama[4] & Toshio Nakaki[1,5 ✉]

Glutathione (GSH) is an important antioxidant that plays a critical role in neuroprotection. GSH depletion in neurons induces oxidative stress and thereby promotes neuronal damage, which in turn is regarded as a hallmark of the early stage of neurodegenerative diseases. The neuronal GSH level is mainly regulated by cysteine transporter EAAC1 and its inhibitor, GTRAP3-18. In this study, we found that the GTRAP3-18 level was increased by up-regulation of the microRNA miR-96-5p, which was found to decrease EAAC1 levels in our previous study. Since the 3'-UTR region of GTRAP3-18 lacks the consensus sequence for miR-96-5p, an unidentified protein should be responsible for the intermediate regulation of GTRAP3-18 expression by miR-96-5p. Here, we discovered that RNA-binding protein NOVA1 functions as an intermediate protein for GTRAP3-18 expression via miR-96-5p. Moreover, we show that intra-arterial injection of a miR-96-5p-inhibiting nucleic acid to living mice by a drug delivery system using microbubbles and ultrasound decreased the level of GTRAP3-18 via NOVA1 and increased the levels of EAAC1 and GSH in the dentate gyrus of the hippocampus. These findings suggest that the delivery of a miR-96-5p inhibitor to the brain would efficiently increase the neuroprotective activity by increasing GSH levels via EAAC1, GTRAP3-18 and NOVA1.

[1] Department of Pharmacology, Teikyo University School of Medicine, Tokyo, Japan. [2] Laboratory of Drug and Gene Delivery, Faculty of Pharma-Science, Teikyo University, Tokyo, Japan. [3] Teikyo University Support Center for Women Physicians and Researchers, Tokyo, Japan. [4] Laboratory of Theranostics, Faculty of Pharma-Science, Teikyo University, Tokyo, Japan. [5] Faculty of Pharma-Science, Teikyo University, Tokyo, Japan. ✉email: nakaki@med.teikyo-u.ac.jp

Neurodegenerative diseases (NDs) such as Alzheimer's disease, Parkinson's disease and multiple system atrophy are chronic and progressive conditions that are caused by a selective loss of neurons in the central nervous system[1]. Although there are several causes of the onset and progression of NDs, a common feature of all these diseases is enhancement of oxidative stress[2], which has been defined as a state in which production of reactive oxygen species (ROS) and reactive nitrogen species (RNS) exceeds the capacity of antioxidant systems to control them[3,4]. Glutathione (GSH) is a major antioxidant that plays crucial roles in the central nervous system, such as by protecting the brain against oxidants[5]. In this regard, the role of GSH compensates for the relatively lower levels of activities of antioxidative enzymes such as superoxide dismutase, catalase and glutathione peroxidase, which are lower in the brain than in the peripheral tissues, while lipids, which are a target of peroxidation, are more abundant in the brain than in other organs[6]. GSH acts as an electron donor and thereby detoxifies either ROS or RNS by oxidizing itself[5]. Abnormalities in the GSH system have been suggested to play an important role in the etiology of various diseases resulting from oxidative stress[7]. In fact, reduced levels of GSH have been observed in the postmortem brains of patients with various NDs[8]. More importantly, GSH levels have been shown to gradually decline from the early stages in the course of ND progression[9,10]. Accumulating clinical evidence obtained using proton magnetic resonance spectroscopy reveals that brain GSH levels are depleted not only in patients with Alzheimer's disease but also those with mild cognitive impairment, which is considered to be an early stage of Alzheimer's disease[10,11]. These studies indicate the importance of early detection of GSH decrease, followed by the initiation of ND treatment.

GSH is composed of three amino acids: cysteine, glutamate and glycine[12]. Among them, cysteine functions as the rate-limiting substrate for neuronal GSH synthesis, and is transported into mature neurons via excitatory amino acid carrier 1 (EAAC1)[13]. Therefore, mice with *EAAC1* deficiency exhibit a marked decrease in neuronal GSH content and increased vulnerability to oxidative stress[13]. Glutamate transporter-associated protein 3-18 (GTRAP3-18), which is an endoplasmic reticulum-localized protein, was originally identified by yeast two-hybrid screening as a protein interacting with the carboxy terminal intracellular domain of EAAC1[14]. The trafficking of EAAC1 from the endoplasmic reticulum to the Golgi is prevented by GTRAP3-18 trapping EAAC1 in the endoplasmic reticulum[15]. Further, mice with *GTRAP3-18* deficiency showed a marked increase in neuronal GSH content and increased neuroprotective activity[15].

MicroRNAs (miRNAs) are small endogenous RNAs that play a posttranscriptional regulatory role in gene expression by targeting mRNAs[16]. Since abnormalities in miRNA expression have been observed in the blood, cerebrospinal fluid and brain in patients with NDs, miRNA dysregulation may be involved in the pathogenesis and pathology of NDs[17]. Our previous study indicated that one miRNA in particular, miR-96-5p, plays a role in the direct regulation of redox states by targeting the 3′-UTR region of EAAC1 mRNA[18]. In addition, we have shown that a miR-96-5p inhibitor exhibited neuroprotective effects by increasing GSH levels, at least in part by increasing EAAC1 expression in the brain[18]. Importantly, a clinical report has shown that the brain expressions of miR-96-5p and its target EAAC1 are deregulated in patients with multiple system atrophy[19], for which no therapeutics are available at present[20]. Considering that dysregulation of miR-96-5p could cause multiple system atrophy through downregulation of EAAC1, miR-96-5p inhibitors would be potential therapeutic agents for multiple system atrophy.

Post-transcriptional gene regulation in eukaryotes is extensively controlled by ribonucleoprotein complexes consisting of RNA-binding proteins (RBPs) and transacting RNAs[21]. Ribonucleoprotein complexes modulate RNA processing, including maturation, stability, transport, editing, and translation of RNA transcripts[22]. Eukaryotic cells encode a large number of RBPs, each of which contains one or more domains with the ability to specifically recognize target transcripts[23]. Neuro-oncological ventral antigen 1 (NOVA1) is an RBP, which was initially identified as an antigen in a rare neurological disorder known as paraneoplastic opsoclonus-myoclonus ataxia[24]. So far, NOVA1 has been reported to be involved in mRNA processing, splicing and miRNA regulation[25,26]. Since NOVA1 is known to be specifically expressed in neurons and tumors, NOVA1 may contribute to the tissue-specific regulation of miRNAs by recruiting miRNA-induced silencing complex via binding to Argonaute proteins[27].

The blood brain barrier (BBB) is a major obstacle to the delivery of therapeutic agents into the brain to treat NDs[28]. The BBB is a strict permeability barrier that is essential for normal brain function—for example, the BBB restricts the entry of xenobiotic substances into the brain to protect brain homeostasis[29]. Recently, a treatment using ultrasound (US) in combination with microbubbles (MBs) has received much attention as a potential technology for delivering drugs and genes[30,31]. The combined use of US and MBs is considered to increase BBB permeability, possibly by disrupting blood vessels, and thereby to enhance the delivery of drugs into neuronal cells in the brain[32,33]. This technology is very useful to deliver nucleic acids and peptides into the brain and may facilitate the treatment of NDs.

In the current study, we show that GTRAP3-18, the negative regulator of EAAC1, is itself indirectly regulated by miR-96-5p. Moreover, the intra-arterial (i.a.) injection of a miR-96-5p inhibitor with MBs in combination with US increased GSH by increasing EAAC1 and decreasing GTRAP3-18 expression in the mouse hippocampus in vivo. These results may have an impact on the development of therapeutics for neuroprotection.

## Results

**Indirect regulation of GTRAP3-18 by miR-96-5p**. In a previous report, we showed that miR-96-5p regulated GSH levels via direct regulation of the cysteine transporter EAAC1 by targeting the 3′-UTR[18]. We speculated that GTRAP3-18, a negative regulator of EAAC1, could also be regulated by miR-96-5p. To examine this working hypothesis, we used a synthesized RNA that had the same sequence as miR-96-5p; this RNA, termed a miR-96-5p mimic, mimics the function of the mature endogenous miR-96-5p.

First, we confirmed the role of miR-96-5p on the GSH level and then investigated its role on GTRAP3-18 expression in a cultured human neuroblastoma cell line, SH-SY5Y. To evaluate the intracellular GSH level, we used 7-amino-4-choloromethylcourmarine (CMAC), which fluoresces upon conjugation with the thiol group of GSH, after transfection of the miR-96-5p mimic and/or its inhibitor. We simultaneously performed an immunocytochemical experiment with an antibody against GTRAP3-18. As shown in Fig. 1a, b, a significantly lower intensity of CMAC was observed in the cells transfected with the miR-96-5p mimic compared to the control ($p = 0.0016$ versus the control by Tukey's HSD test, $d = -3.74$, 95% CI: $-4.91$ to $-2.31$), while the expression of GTRAP3-18 appeared to be increased following transfection of the miR-96-5p mimic (Fig. 1a and Supplementary Fig. 1). Furthermore, the effect on the CMAC level was significantly blocked by co-transfection of the miR-96-5p mimic and inhibitor ($p = 0.012$ versus the effect of the miR-96-5p mimic alone by Tukey's HSD test, $d = 1.35$, 95% CI: 0.38 to 2.22) (Fig. 1b). This inhibitory effect was also observed in the

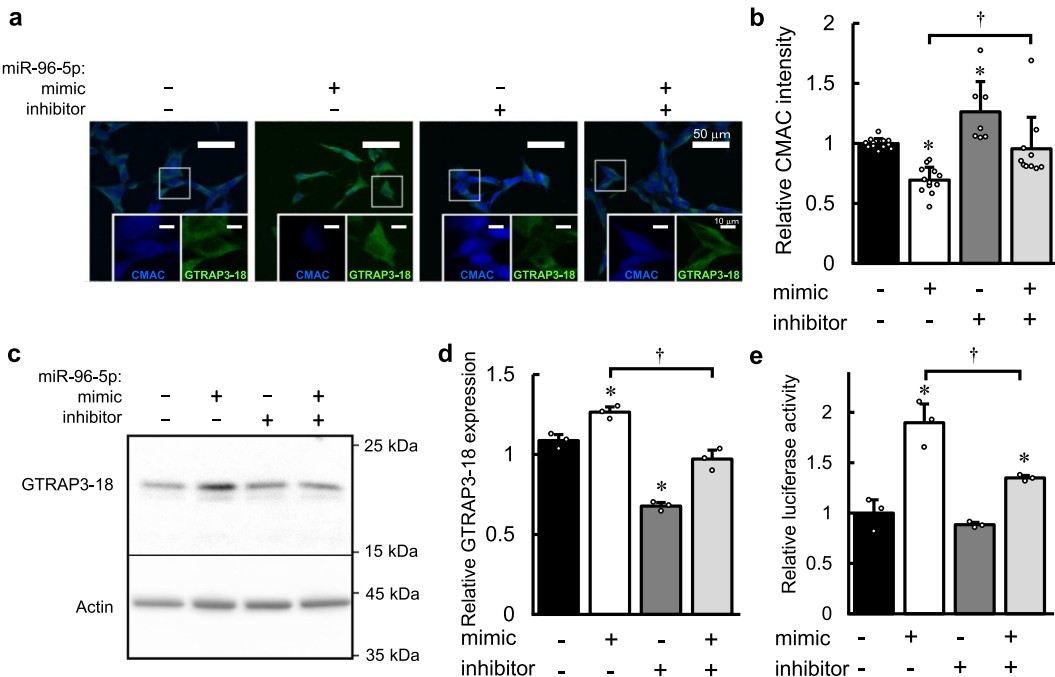

**Fig. 1 The effect of miR-96-5p on GSH level and GTRAP3-18 expression. a** Confocal images show the effect of the miR-96-5p mimic and/or inhibitor transfection on the intensity of CMAC as a marker of GSH (blue) and the expression of GTRAP3-18 (green). Scale bar, 50 μm. Insets in panels are enlarged images corresponding to the box highlighted on the full images. Scale bar, 10 μm. **b** Quantification of the relative CMAC density in **a** is shown. Data are mean values ± SD obtained from independent samples and individual data points are plotted ($n = 12$ for negative control or miR-96-5p mimic, $n = 7$ for miR-96-5p inhibitor and $n = 10$ for both miR-96-5p mimic and inhibitor transfection). Data were analyzed by one-way ANOVA ($F(3,37) = 14.345$, $p = 0.0000023$) and Tukey's HSD test. *$p < 0.05$ relative to the negative control. †$p < 0.05$ versus the effect of the miR-96-5p inhibitor. **c** The endogenous expressions of GTRAP3-18 and β-actin in Neuro2a cells with transfection of miR-96-5p mimic and/or inhibitor are shown. Molecular weight markers are depicted at right. Full blots are presented in Supplementary Fig. 12. **d** Quantification of the data in panel **c** by densitometry. Data represent mean values ± SD obtained from three independent experiments and individual data points are plotted. Data were analyzed by one-way ANOVA ($F(3,8)=79.74$, $p = 0.0000027$) and Tukey's HSD test. *$p < 0.05$ relative to the negative control. †$p < 0.05$ versus the effect of the miR-96-5p inhibitor. **e** Relative luciferase activity in SH-SY5Y cells transfected with the luciferase plasmid of GTRAP3-18 3′-UTR with the miR-96-5p mimic and/or inhibitor is shown. Data represent mean values ± SD obtained from three individual experiments and individual data points are plotted. Data were analyzed by one-way ANOVA($F(3,8) = 31.27$, $p = 0.000091$) and Tukey's HSD test. *$p < 0.05$ relative to the negative control. †$p < 0.05$ versus the effect of the miR-96-5p inhibitor.

intensity of GTRAP3-18 immunostaining (Fig. 1a and Supplementary Fig. 1). Moreover, the transfection of the miR-96-5p inhibitor alone resulted in a significantly higher intensity of CMAC ($p = 0.025$ versus the control by Tukey's HSD test, $d = 1.73$, 95% CI: 0.58 to 2.73) and decreased expression of GTRAP3-18 compared to the control (Fig. 1a, b and Supplementary Fig. 1). These results suggest that miR-96-5p has effects on the expression of GTRAP3-18 as well as on the intracellular GSH level.

We further examined the role of miR-96-5p on the ROS level using the mouse neuroblastoma cell line Neuro2a. We performed a ROS measurement assay after hydrogen peroxide ($H_2O_2$) treatment of cells transfected with an appropriate combination of the miR-96-5p mimic and inhibitor. The results showed that the intracellular ROS level was increased in the cells transfected with the miR-96-5p mimic compared to the control when the cells were treated with 500 μM $H_2O_2$ ($p = 0.013$ versus the control by Tukey's HSD test, $d = 2.01$, 95% CI: 0.50 to 3.21) (Supplementary Fig. 2). In addition, the miR-96-5p inhibitor decreased the intracellular ROS level compared to that in the control when the cells were treated with 100 μM $H_2O_2$ ($p = 0.041$ versus the control by Tukey's HSD test, $d = -2.21$, 95% CI: -3.43 to -0.64) (Supplementary Fig. 2). These results suggested that overexpression of miR-96-5p induced an increase in the ROS level, while miR-96-5p inhibition reduced oxidative stress.

Next, we investigated the role of miR-96-5p on GTRAP3-18 expression using a western blotting method that provides more quantitative results than immunocytochemistry. We used

Neuro2a cells for the transfection experiments because of their high transfection efficiency[34] and sufficiently high expression of endogenous GTRAP3-18. The results showed that the expression of the GTRAP3-18 protein was increased 1.26-fold by transfection of the miR-96-5p mimic compared to the transfection of a negative control ($p = 0.0081$ versus the control by Tukey's HSD test, $d = 5.03$, 95% CI: 1.23 to 6.79) (Fig. 1c, d). Next, we transfected a miR-96-5p inhibitor with the miR-96-5p mimic to examine their combined effect on the GTRAP3-18 expression. The protein expression of GTRAP3-18 was lower by this combined treatment than by administration of the miR-96-5p mimic alone, indicating that the effect of the miR-96-5p mimic was blocked by the miR-96-5p inhibitor ($p = 0.0032$ versus the effect of the combined treatment by Tukey's HSD test, $d = -6.44$, 95% CI: -8.45 to -1.82) (Fig. 1d). Moreover, the expression of the GTRAP3-18 protein was lower by the transfection of the miR-96-5p inhibitor alone compared to the negative control ($p = 0.000028$ versus the control by Tukey's HSD test, $d = -13.10$, 95%CI: -16.58 to -4.32) (Fig. 1d). Similar results were obtained using cells of the human embryonic kidney cell line HEK293 (Supplementary Fig. 3). In addition, we performed quantitative RT-PCR to detect the mRNA level of GTRAP3-18, and the results showed a similar tendency, albeit not a statistically significant one (Supplementary Fig. 4). These results were totally unexpected, because miRNAs generally function as negative regulators of target genes by direct binding of their 3′-UTRs[35]. The regulatory mechanism of GTRAP3-18 by miR-96-5p is

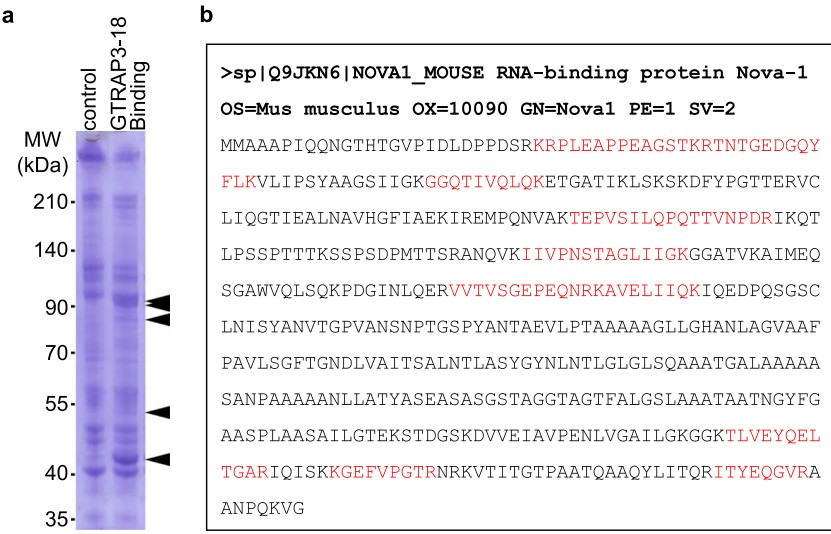

**Fig. 2 Identification of the protein mediating the increase of GTRAP3-18 expression by miR-96-5p. a** The binding protein samples extracted from the magnetic beads conjugated with GTRAP3-18 3′-UTR RNA or scrambled RNA (control) were loaded on the SDS-PAGE gel. CBB stained gels are shown. Arrows indicate the bands analyzed by mass spectrometry. Molecular weight markers are depicted at left. The full-length gel is presented in Supplementary Fig. 13. **b** A result of mass spectrometry analysis of the purified protein with the matched amino acids is shown (red). List of the RBP candidates is presented in Supplementary Table 1.

expected to be completely different from that of EAAC1 and, indeed, unique. The increase in target gene expression by miRNA indicates the presence of some intermediates.

To examine whether miR-96-5p regulates GTRAP3-18 via the 3′-UTR either directly or indirectly, we made a construct of GTRAP3-18 3′-UTR cloned into a luciferase reporter plasmid and performed a luciferase reporter gene assay. Consistent with the result of the western blotting analysis that the endogenous expression of GTRAP3-18 was increased by the miR-96-5p mimic, luciferase activity was significantly higher when the miR-96-5p mimic was transfected than in the negative control ($p = 0.00024$ versus the control by Tukey's HSD test, $d = 5.58$, 95% CI: 1.47 to 7.43) (Fig. 1e). This effect was blocked by the miR-96-5p inhibitor, which was again consistent with the western blotting analysis ($p = 0.0062$ versus the effect of inhibitor by Tukey's HSD test, $d = -4.13$, 95% CI: $-5.75$ to $-0.84$) (Fig. 1d, e).

To confirm that the 3′-UTR sequence of GTRAP3-18 has target sites of miR-96-5p, we performed a computational analysis using several miRNA prediction databases, and determined that there were no target sites of miR-96-5p on the 3′-UTR of GTRAP3-18[36]. These results would seem to largely confirm that GTRAP3-18 is regulated by miR-96-5p through some other factors affecting the 3′-UTR of GTRAP3-18, possibly RBPs.

**Screening of RNA-binding proteins which mediate up-regulation of GTRAP3-18 by miR-96-5p.** To identify the factors mediating the effect of miR-96-5p on the GTRAP3-18 protein expression, we performed a pull-down assay using in vitro transcribed RNA containing the GTRAP3-18 3′-UTR sequence conjugated to magnetic beads. After the mouse brain lysate was mixed with the beads, the binding proteins were separated and stained. Compared with the negative control using scrambled RNA, several bands appeared on the lane loaded with the extracted binding proteins of the GTRAP3-18 3′-UTR sequence conjugating beads (Fig. 2a). Among them, five bands which showed a more than 2-fold increase in the lane for GTRAP3-18 binding compared to the control were selected as candidate binding proteins for further analysis. These bands were then analyzed using a nano LC-MS/MS system and identified using the MASCOT MS/MS ion search method by Shimadzu Techno

Research Inc. As a result of this analysis, we obtained several candidates for binding proteins of GTRAP3-18 3′-UTR (Supplementary Table 1). To narrow the list of candidates, we established a criterion: the protein that mediates the effect of miR-96-5p on GTRAP3-18 must be regulated by miR-96-5p. For this reason, the top-ranking protein possessing target sites of miR-96-5p in several miRNA databases[36] was chosen as a candidate regulatory protein of GTRAP3-18; this protein was NOVA1 (Fig. 2b). NOVA1 is an RBP, and its mRNA was originally reported to be expressed in tumors and neurons[24]. It has been considered a target antigen in a human paraneoplastic motor disorder[37], and it has also been described as a regulator of RNA splicing in a specific subset of developing neurons and a regulator of neuronal miRNA function[27,38].

**Direct regulation of NOVA1 expression by miR-96-5p.** To confirm that NOVA1 is regulated by miR-96-5p, we transfected the miR-96-5p mimic and/or inhibitor and detected the protein expression of NOVA1 in Neuro2a cells. Western blotting analysis revealed that NOVA1 expression was decreased by the transfection of the miR-96-5p mimic ($p = 0.00080$ versus the control by Tukey's HSD test, $d = -3.44$, 95% CI: $-4.94$ to $-1.26$), and this effect was blocked by co-transfection of the miR-96-5p mimic and inhibitor, as expected ($p = 0.00021$ versus the effect of inhibitor by Tukey's HSD test, $d = 3.42$, 95% CI: 1.25 to 4.92) (Fig. 3a, b). Next, to examine whether the effect of miR-96-5p on the NOVA1 expression is a direct one, we performed a luciferase reporter gene assay using a construct of NOVA1 3′-UTR cloned into a luciferase reporter plasmid. Consistent with the results of western blotting, the level of luciferase activity was significantly lower when the miR-96-5p mimic was transfected compared to the negative control ($p = 0.00077$ versus the control by Tukey's HSD test, $d = -3.65$, 95% CI: $-5.00$ to $-1.90$) (Fig. 3c). This reduction was again blocked by the miR-96-5p inhibitor ($p = 0.031$ versus the effect of inhibitor by Tukey's HSD test, $d = 0.88$, 95% CI: 0.43 to 2.06) (Fig. 3c).

We next tried to identify the target sites of NOVA1 by miR-96-5p using several miRNA algorithms[36]. We found two target sites of miR-96-5p on the 3′-UTR of NOVA1 (Fig. 3d). We made luciferase reporter plasmids of NOVA1 3′-UTR constructs with

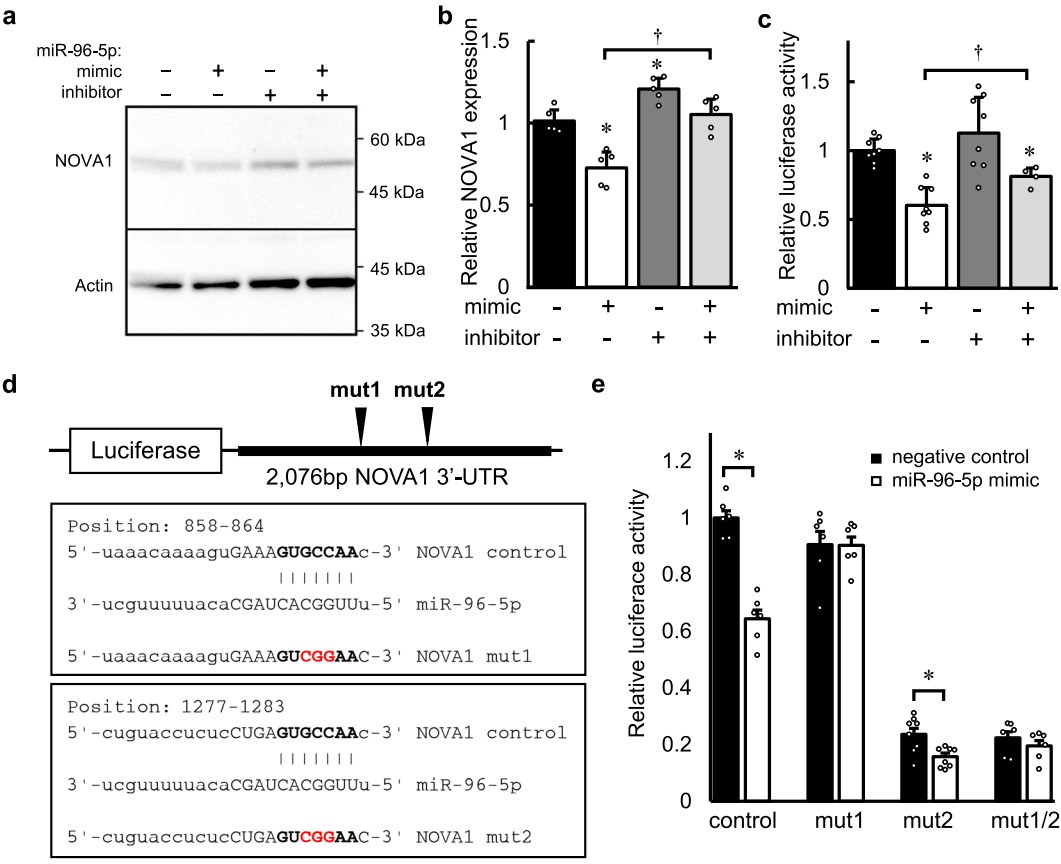

**Fig. 3 The effect of miR-96-5p on the expression of an RNA-binding protein, NOVA1. a** The endogenous expressions of NOVA1 and β-actin in Neuro2a cells with transfection of the miR-96-5p mimic and/or inhibitor are shown. Molecular weight markers are depicted at right. The full blots are presented in Supplementary Fig. 14. **b** Densitometric quantification of the data in panel **a** is shown. Data represent mean values ± SD obtained from five independent experiments and individual data points are plotted. Data were analyzed by one-way ANOVA($F(3,16) = 23.84$, $p = 0.0000038$) and Tukey's HSD test. *$p < 0.05$ relative to the negative control. †$p < 0.05$ versus the effect of the miR-96-5p inhibitor. **c** Relative luciferase activity in SH-SY5Y cells transfected with the luciferase plasmid of NOVA1 3′-UTR with the miR-96-5p mimic and/or inhibitor are shown. Data represent mean values ± SD obtained from independent samples ($n = 8$ for negative control, miR-96-5p mimic or miR-96-5p inhibitor, $n = 4$ for miR-96-5p mimic plus inhibitor) and individual data points are plotted. Data were analyzed by one-way ANOVA($F(3,24) = 13.24$, $p = 0.000027$) and Tukey's HSD test. *$p < 0.05$ relative to the negative control. †$p < 0.05$ versus the effect of the miR-96-5p inhibitor. **d** A schematic plot of the luciferase constructs of NOVA1 3′-UTR is shown. The sequences for the predicted miR-96-5p target sites are shown in the box. Mutation was added in a core sequence (red font) of the miR-96-5p target (bold font). **e** Relative luciferase activities in SH-SY5Y cells transfected with the luciferase plasmids in **d** with the miR-96-5p mimic are shown. Data represent mean values ± SD obtained from independent samples ($n = 6$ for control, mut1 or mut1/2, $n = 8$ for mut2) and individual data points are plotted. Data were analyzed by Student's *t*-test, two-sided. *$p < 0.05$ relative to the negative control in each construct.

point mutation(s) of miR-96-5p target sites and performed a luciferase reporter gene assay. We found that one of the mutations on the NOVA1 3′-UTR target site (mut1) blocked the reduction of luciferase activity by the transfection of miR-96-5p mimic ($t(10) = -0.052$, $p = 0.959$ by Student's *t*-test, two-sided, $d = -0.033$, 95% CI: −1.16 to 1.10), which was consistent with the bioinformatic prediction (Fig. 3e). In contrast, when a luciferase plasmid with a mutation at another site (mut2) was used, luciferase activity was reduced by transfection of the miR-96-5p mimic, as was the luciferase activity of the non-mutation plasmid ($t(14) = -2.953$, $p = 0.027$ by Student's *t*-test, two-sided, $d = -1.58$, 95% CI −2.60 to −0.38), which was not consistent with the bioinformatic prediction. These results indicate that the target site of miR-96-5p is located in the region at positions 858 to 864 of NOVA1 3′-UTR.

**Regulation of GTRAP3-18 by NOVA1.** To clarify whether the expression of GTRAP3-18 is affected by the expression level of NOVA1, we knocked down the NOVA1 expression using its specific siRNA. As shown by the results of western blotting

analysis in Fig. 4a, b, we achieved 73.5% knockdown of NOVA1 expression in the Neuro2a cells ($t(10) = -5.037$, $p = 0.00051$ by Student's *t*-test, two-sided, $d = -3.76$, 95% CI: −5.27 to −1.68). Further, the analysis of cells transfected with NOVA1 siRNA showed that knockdown of NOVA1 resulted in a 1.85-fold increase in the protein level of GTRAP3-18 compared to the transfection of scrambled siRNA as a negative control ($t(4) = 12.88$, $p = 0.00021$ by Student's *t*-test, two-sided, $d = 12.88$, 95% CI: 4.25–16.31) (Fig. 4a, c). Similar results were obtained using HEK293 cells (Supplementary Fig. 5). In addition, immunocytochemical analysis using SH-SY5Y cells also showed that the intensity of GTRAP3-18 expression was presumably increased by NOVA1 knockdown (Supplementary Fig. 5). These results were consistent with the findings of the western blotting and immunocytochemical analysis that the expressions of NOVA1 and GTRAP3-18 were reduced and increased, respectively, by transfection of the miR-96-5p mimic, suggesting that NOVA1 downregulates the expression of GTRAP3-18, leading to an increase in the GSH level in cultured cells.

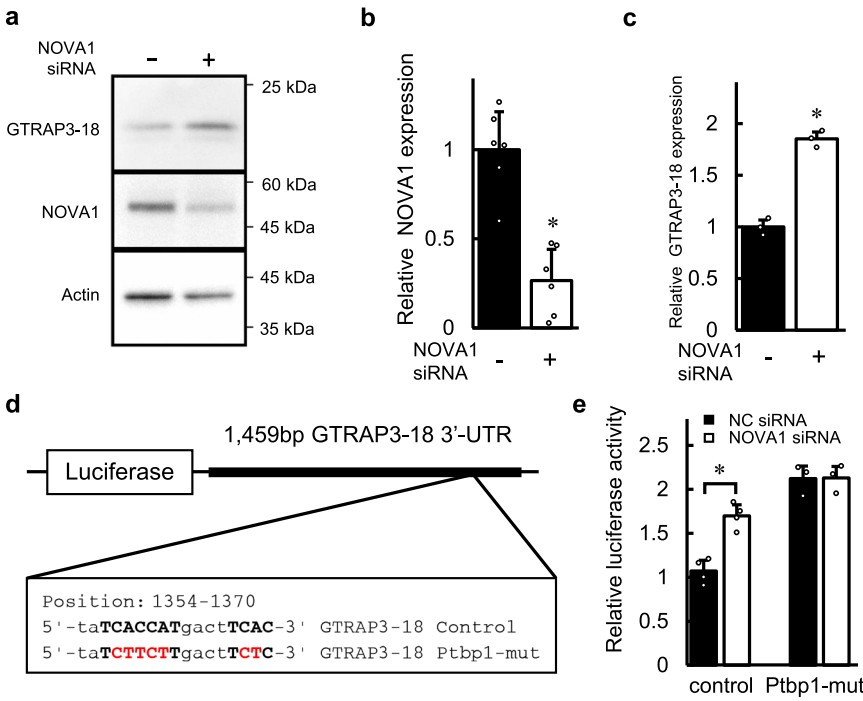

**Fig. 4 The effect of NOVA1 knockdown on the expression of GTRAP3-18. a** The endogenous expressions of GTRAP3-18, NOVA1 and β-actin after transfection of NOVA1 siRNA (+) or negative control siRNA (−) are shown. Molecular weight markers are depicted at right. Full blots are presented in Supplementary Fig. 15. **b** Densitometric quantification of the NOVA1 expression in **a** is shown. Data represent mean values ± SD obtained from six individual experiments and individual data points are plotted. Data were analyzed Student's $t$-test, two-sided. *$p < 0.05$ relative to the negative control. **c** Densitometric quantification of the GTRAP3-18 expression in **a** is shown. Data represent mean values ± SD obtained from three individual experiments and individual data points are plotted. Data were analyzed by Student's $t$-test, two-sided. *$p < 0.05$ relative to the negative control. **d** A schematic plot of the luciferase constructs of GTRAP3-18 3′-UTR is shown. The sequence for the predicted NOVA1-binding site is shown in the box. Mutation was added in a core sequence (red font) of tandem YCAY clusters (bold font). The mutated sequence contains a predicted binding site of Ptbp1 (Ptbp1-mut). **e** Relative luciferase activities in SH-SY5Y cells transfected with the luciferase plasmids in **d** with the NOVA1 siRNA or negative control (NC) are shown. Data represent mean values ± SD obtained from independent samples ($n = 4$ for control, $n = 3$ for Ptbp1-mut) and individual data points are plotted. Data were analyzed by Student's $t$-test, two-sided. *$p < 0.05$ relative to the negative control.

NOVA1 is an RBP that is known as a brain-specific molecule regulating neuronal alternative splicing[24]. Interestingly, NOVA1 was recently reported to be a component of the neuronal miRNA-induced silencing complex[27]. NOVA1 contains hnRNP K homology domains and binds RNA at YCAY (Y = C or T) clusters[39]. A search of the RNA-binding protein database revealed a non-canonical tandem repeat of YCAY in the sequence of GTRAP3-18 3′-UTR (Fig. 4d)[40]. To examine whether NOVA1 regulation is mediated via NOVA1 binding to YCAY clusters on the GTRAP3-18 3′-UTR, we made a luciferase reporter construct of GTRAP3-18 3′-UTR with an inserted mutation of YCAY clusters (Fig. 4d). This mutant contains a consensus sequence of the Polypyrimidine tract-binding protein 1 (Ptbp1)-binding site, which is a CUU repeat. As mentioned above, the luciferase activity of the GTRAP3-18 3′-UTR reporter without any mutations is increased when NOVA1 siRNA is transfected ($t(6) = 6.20$, $p = 0.00081$ by Student's $t$-test, two-sided, $d = 5.06$, 95% CI: 1.83 to 6.96) (Fig. 4e). In contrast, this mutant of YCAY clusters (Ptbp1-mut) blocked the increase of luciferase activity by the transfection of NOVA1 siRNA, which was consistent with the bioinformatic prediction ($t(4) = 0.054$, $p = 0.959$ by Student's $t$-test, two-sided, $d = 0.054$, 95% CI: −1.56 to 1.64) (Fig. 4e). Moreover, this mutant also blocked the increase of luciferase activity by the transfection of the miR-96-5p mimic ($p = 0.657$ by Tukey's HSD test, $d = -1.51$, 95% CI: −2.95 to 0.53) (Supplementary Fig. 6). We also made luciferase reporter plasmids of GTRAP3-18 3′-UTR constructs with insertion or deletion mutation of YCAY clusters, which are expected not to be bound

with proteins including NOVA1. These constructs also blocked the increase of luciferase activity by transfection of the miR-96-5p mimic. (Supplementary Fig. 7). These results indicate that the regulation of GTRAP3-18 expression by miR-96-5p is mediated by binding of NOVA1 on the 3′-UTR of GTRAP3-18. Interestingly, the level of luciferase activity of the Ptbp1 mutant of GTRAP3-18 3′-UTR was more than 2-fold higher compared to that of the construct without mutations, which is consistent with an inhibitory effect of NOVA1 on GTRAP3-18 expression. Further, the levels of luciferase activity of NOVA1 unbound mutants of GTRAP3-18 3′-UTR constructs were much lower than that of the construct without mutations (Supplementary Fig. 7). These results suggest that protein binding in this region of YCAY clusters is important for the stability of the GTRAP3-18 mRNA or protein.

**Intra-arterial administration of a miR-96-5p inhibitor using ultrasound sonication with microbubbles.** To determine whether miR-96-5p regulates the GSH level via GTRAP3-18 and NOVA1 in vivo, we performed i.a. injection of a miR-96-5p inhibitor to block endogenous miR-96-5p miRNA function in the brain. As shown in Fig. 5a, during the surgical procedure of i.a. injection, the catheter was inserted to the left common carotid artery and reached towards the internal carotid artery. The left side of the brain is regarded as the treated side and the right side as the non-treated side. For the drug delivery, drug and MBs were mixed and then administered concomitant with US sonication.

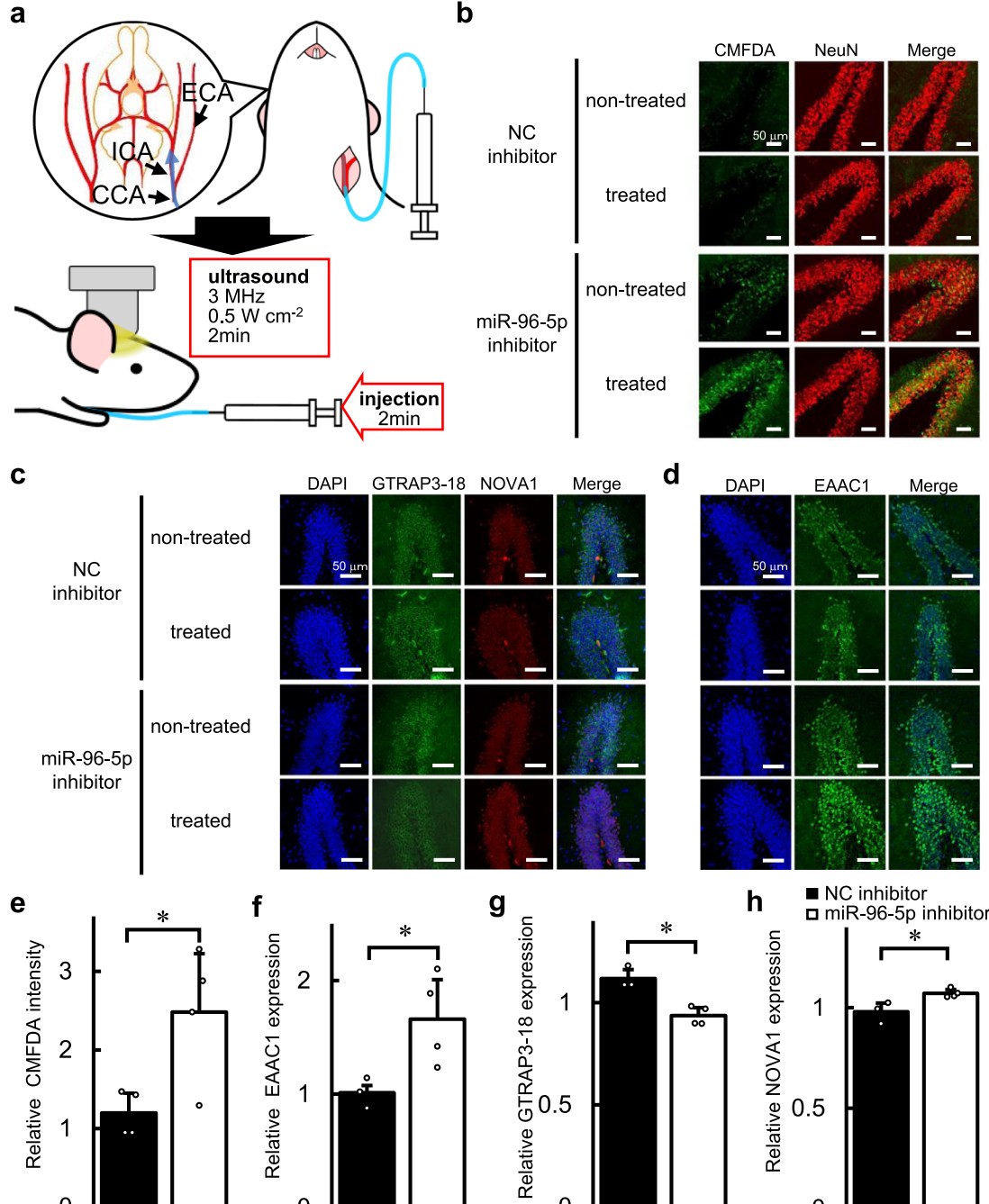

**Fig. 5 Effect of the intra-arterial administration of miR-96-5p inhibitor with MBs and US on the levels of GSH, EAAC1, GTRAP3-18, and NOVA1.**
**a** A schematic of the intra-arterial (i.a.) injection is shown. The common carotid artery (CCA), external carotid artery (ECA), and internal carotid artery (ICA) are exposed, and a small incision is made on the CCA. The catheter is inserted and advanced towards the ICA; once the catheter is stably positioned, a mixture of miR-96-5p inhibitor and MBs is injected with a programmed microinjector. Simultaneously with the injection, ultrasound (US) is administered from a probe placed on the shaved heads of the mice. **b** Confocal images show the effect of administration of a miR-96-5p inhibitor or a negative control (NC) inhibitor on the intensity of CMFDA as an intracellular GSH marker (green). NeuN is stained as a neuronal nuclear marker (red) and co-localized with the CMFDA signal. Scale bar, 50 μm. **c** Confocal images show the effect of administration of a miR-96-5p inhibitor or an NC inhibitor on the expressions of GTRAP3-18 (green) and NOVA1 (red). The nuclei were stained with DAPI (blue). Scale bar, 50 μm. **d** Confocal images show the effect of administration of a miR-96-5p inhibitor or an NC inhibitor on the EAAC1 expression (green). The nuclei were stained with DAPI (blue). Scale bar, 50 μm. **e** The level of CMFDA intensity after the administration of NC or miR-96-5p inhibitor are shown. Data are mean values ± SD obtained from four independent experiments and individual data points are plotted. Data were analyzed by Student's *t*-test, two-sided. *p < 0.05 relative to NC inhibitor. **f–h** The expression of EAAC1 (**f**), GTRAP3-18 (**g**), and NOVA1 (**h**) after the administration of NC or miR-96-5p inhibitor are shown. Data are mean values ± SD obtained from independent samples (n = 3 for NC inhibitor, n = 4 for miR-96-5p inhibitor) and individual data points are plotted. Data were analyzed by Student's *t*-test, two-sided. *p < 0.05 relative to NC inhibitor.

First, we confirmed the BBB leakage using Evans blue staining on the treated side of the hippocampus (Supplementary Fig. 8). Signals corresponding to the Evans blue dye were detected in a limited area around the hippocampus, and were especially strong in the dentate gyrus (DG) of the hippocampus. Next, we performed an i.a. injection of miR-96-5p inhibitor with MBs and applied US. Six days after the administration of the miR-96-5p inhibitor, slice culture experiments were performed to detect the amount of GSH in the DG. To evaluate the amount of GSH in the brain, we used 5-chloromethylfluorescein diacetate (CMFDA), which is a cell-permeant green fluorescent probe that reacts with thiol-group substances, mainly GSH, in the brain. We compared the CMFDA signals as GSH content between the treated and non-treated side of the DG. The CMFDA signal in the DG treated with the miR-96-5p inhibitor increased 2.48-fold as compared to that on the other side, whereas, when using the negative control inhibitor, the changes in CMFDA intensity were small between the treated side and the other side of the DG ($t(6) = 2.84$, $p = 0.030$ by Student's $t$-test, two-sided, $d = 2.32$, 95% CI: 0.31 to 3.71) (Fig. 5b, e). Next, we performed immunohistochemical analysis to detect the expression level of GTRAP3-18 and NOVA1 in the DG. We found that the ratio of GTRAP3-18 expression on the side treated with miR-96-5p inhibitor to GTRAP3-18 expression on the non-treated side was 0.87-fold lower than the ratio of GTRAP3-18 expression treated with negative control inhibitor ($t(5) = -4.87$, $p = 0.0046$ by Student's $t$-test, two-sided, $d = -4.42$, 95% CI: $-6.17$ to $-1.26$) (Fig. 5c, g). In addition, the ratio of NOVA1 protein level on the side treated with the miR-96-5p inhibitor to non-treated side was 1.14-fold higher than the ratio of NOVA1 level treated with negative control inhibitor ($t(5) = 3.18$, $p = 0.024$, by Student's $t$-test, two-sided, $d = 2.94$, 95% CI: 0.49 to 4.45) (Fig. 5c, h). We also performed the same experiment to detect the expression of EAAC1 in the DG, and found that EAAC1 was significantly increased by 1.69-fold on the treated side as compared to the non-treated side when the mice were injected with the miR-96-5p inhibitor, while the changes were small by the negative control administration ($t(5) = 2.63$, $p = 0.046$ by Student's $t$-test, two-sided, $d = 2.33$, 95% CI: 0.14 to 3.77) (Fig. 5d, f). The change in EAAC1 expression by injection of the miR-96-5p inhibitor was larger than the changes in GTRAP3-18 and NOVA1 expression. Since EAAC1 is regulated by GTRAP3-18, treatment with a miR-96-5p inhibitor may yield a synergistic effect on EAAC1 expression by the dual control of GTRAP3-18 and EAAC1 itself. These results suggest that the miR-96-5p inhibitor increased the neuroprotective GSH by downregulating GTRAP3-18 via NOVA1 as well as by upregulating EAAC1. Further, the drug delivery method using MBs with US application is useful for increasing GSH in neuronal cells in the DG of the mouse brain.

We performed additional immunohistochemical experiments to detect microglial and glial activation using anti-Iba1 and anti-GFAP antibodies, respectively, since it has been reported that US could lead to microglial and glial activation (Supplementary Fig. 9). The results showed that Iba1 immunostaining as a microglial marker was clearly increased with the US application compared to the sham operation (Supplementary Fig. 9). It is of interest that microglial activation was reduced with the administration of a miR-96-5p inhibitor compared to the negative control (Supplementary Fig. 9). However, there were no differences in GFAP immunostaining as a glial marker between sham operation and US application with injection of MBs and miRNA inhibitors (Supplementary Fig. 9). In addition, the immunostaining of 4-HNE as a lipid oxidation marker was similarly unaffected by any of the treatments (Supplementary Fig. 10). We also performed immunohistochemical experiments to detect autophagy activation using an antibody against LC3, because autophagy dysfunction has been implicated in the pathogenesis of several NDs. Interestingly, LC3 immunostaining as an autophagy marker was increased in the DG treated with the miR-96-5p inhibitor (Supplementary Fig. 10). These results suggest that the miR-96-5p inhibitor had a neuroprotective effect through an unknown mechanism other than GSH regulation.

## Discussion

To our best knowledge, this is the first study to show that a miRNA enhanced the function of a target protein in a binary and synergistic manner—i.e., by directly stimulating the expression of the target protein as well as by inhibiting the expression of a protein with negative effects on the target function. This is a cutting-edge concept for regulating homeostasis. We have shown in this study that an excess of miR-96-5p expression decreases GSH levels through downregulation of EAAC1, in addition to upregulation of GTRAP3-18 by NOVA1 reduction.

In our previous study, we showed that EAAC1 is directly regulated by miR-96-5p, and intracerebroventricular injection of a miR-96-5p inhibitor potentiates the neuroprotective activity by increasing the GSH level via EAAC1 in the brain[18]. Clearly, however, intracerebroventricular injection is not an appropriate route for administrating therapeutics in humans. In this study, we further showed that GTRAP3-18, a negative regulator of EAAC1, is also indirectly regulated by miR-96-5p. We obtained several candidate proteins which could mediate the effect of miR-96-5p on GTRAP3-18 expression from brain extract using an RNA-protein binding assay. Among them, NOVA1 was the most likely candidate that satisfied the criteria, exhibiting both direct binding to the 3′-UTR of GTRAP3-18 and direct regulation by miR-96-5p. NOVA1 is one of the RBPs harboring a hnRNP K homology domain[41], which is a predicted binding site on the 3′-UTR of GTRAP3-18. In fact, we showed that NOVA1 expression was decreased via the direct regulation of miR-96-5p; NOVA1 directly bound to YCAY clusters on the 3′-UTR of GTRAP3-18; and finally, the expression of GTRAP3-18 increased. These results raise the prospect of a model of the regulation of gene expression by miRNA. In this model, a miRNA regulates a target protein expression via direct and indirect pathways.

NOVA1 was initially identified as an antigen in a rare neurological disorder known as paraneoplastic opsoclonus-myoclonus ataxia[24]. In subsequent studies, NOVA1 was recognized as an RBP involved in miRNA regulation as well as mRNA processing and splicing[25,26]. Recently, Storchel et al. reported that NOVA1 physically interacts with Argonaute proteins, which are core components of the miRNA-induced silencing complex[27]. Interestingly, one of the well-known RBPs, Pumilio (PUM), binds to the 3′-UTR of target genes and induces conformational changes in the RNA structure that favor association with miRNAs[42]. Moreover, the PUM-binding site is located close to miRNA-targeting sites, and together the PUM-binding site and miRNA-targeting sites could form a stem-loop sequence[42]. Surprisingly, analysis of RNA-fold predictions (RNAfold Webserver[43]) shows that the predicted NOVA1-binding site on GTRAP3-18 3′-UTR is a stem-loop structure[44], and that the stem sequence is a target site of miRNAs such as miR-591 and miR-6502-3p[36] (Supplementary Fig. 11), suggesting that NOVA1 would have a similar role as PUM for regulating GTRAP3-18 expression. Further research is required to identify the precise regulatory mechanism of NOVA1 on GTRAP3-18. Furthermore, NOVA1 was originally recognized as a brain-specific molecule that is highly expressed in the brain, implying that the miRNA regulation of GTRAP3-18 induced by NOVA1 could be brain-specific. Tissue-specific expression of RBPs might play a role in conferring tissue specificity to miRNA regulation.

The search for suitable strategies for brain drug delivery presents a challenge in ND therapeutics because of the BBB[45]. MB technologies could be one of the solutions for drug delivery to the brain[46]. In this study, we administered a miR-96-5p inhibitor with MBs via the internal carotid artery and then administered US sonication from probes placed on the heads of mice. The effect of the miR-96-5p inhibitor was particularly prominent in the DG of the hippocampus, possibly because of the limited US exposure area. The level of GSH was significantly increased on the side with i.a. injection of the miR-96-5p inhibitor, along with an increase in EAAC1 expression and decrease in GTRAP3-18 expression. We thus succeeded in delivering the miR-96-5p inhibitor under somewhat limited conditions, but further research is needed to solve the problem of delivering a miR-96-5p inhibitor to a limited brain area. In addition, i.a. administration requires advanced surgery, which is generally inappropriate for clinical treatment. Improvement and modification of MB techniques will be needed before intravenous or retro-orbital injection can be used in place of i.a. administration from the internal carotid artery[47,48].

There have been reports showing that US itself induces oxidative stress and inflammatory responses with microglial and glial activation[49,50]. In this study, microglial activation was observed with the application of US and MBs, and this activation was surprisingly inhibited by the simultaneous administration of a miR-96-5p inhibitor. In addition, an autophagy system, dysfunction of which has been implicated in the pathogenesis of NDs, was also promoted by miR-96-5p inhibitor administration. Although these results indicate another neuroprotective pathway of the miR-96-5p inhibitor, further investigations will be needed to elucidate this pathway.

In summary, we have shown that miR-96-5p plays an important role in the maintenance of the GSH level in two ways: indirectly via GTRAP3-18 in a manner mediated by NOVA1, and directly by regulating EAAC1. miR-96-5p inhibitors are potential therapeutic agents for multiple system atrophy and other NDs, since they could increase the neuronal GSH level before neuronal cell death occurs in these disorders. The difficulty in the delivery of nucleic acids, including miR-96-5p inhibitors, to the brain could be solved by using MBs with US to cross the BBB.

## Methods

**Cell culture**. SH-SY5Y (kindly gifted by Dr. Shinichi Kohsaka, National Institute of Neuroscience in 2002) was grown in Dulbecco's modified Eagle's medium (Sigma-Aldrich) supplemented with 10% fetal bovine serum (Gibco) and 1% penicillin–streptomycin (Gibco). SH-SY5Y cells were passaged every two to three days with passage range from $5 \times 10^5$ to $1 \times 10^6$ cells per 100-mm dish. Neuro2a (purchased from KAC Co., Ltd.) was grown in Eagle's minimum essential medium (Sigma-Aldrich) supplemented with 10% fetal bovine serum (Gibco), non-essential amino acids (Wako) and 1% penicillin–streptomycin (Gibco). Neuro2a cells were passaged every two to three days with passage range from $1 \times 10^5$ to $3 \times 10^5$ cells per 100-mm dish. HEK293 cells (purchased from DS Pharma Biomedical Co., Ltd.) were grown in minimum essential medium (Gibco) supplemented with 10% fetal bovine serum (Gibco) and 1% penicillin–streptomycin (Gibco). HEK293 cells were passaged every two to three days with passage range from $1 \times 10^5$ to $5 \times 10^5$ cells per 100-mm dish. The cultured cells were maintained at 37 °C in a 5% $CO_2$ humidified incubator (Panasonic). The cells were authenticated by checking morphology using microscope and growth curve analysis. All cell lines were tested by checking with DAPI staining and were found to be mycoplasma-free.

**Transfection**. SH-SY5Y and HEK293 cells were transfected with an siRNA specific for NOVA1, a scrambled siRNA as its negative control (Ambion) or an appropriate combination of a miR-96-5p mimic, a miR-96-5p inhibitor and their negative controls (Exiqon) using Lipofectamine RNAiMax (Invitrogen) according to the manufacturer's protocol as follows. For the analysis of western blotting and immunocytochemistry, siRNA (15 pmol) or appropriate combination of miRNA mimic (30 pmol) and miRNA inhibitor (90 pmol) was diluted in 50 μL of appropriated medium without serum in the well of the 12-well culture plate. Then diluted Lipofectamine RNAiMAX (2.5 μL) in 50 μL of the serum-free medium was added to each well and incubated for 10–20 min at room temperature. In the meantime, cells were diluted in growth medium containing the appropriate

number of cells to give 30–50% confluence within 24 h after plating. Then 1 mL of diluted cells were added to each well. After cells were incubated approximately 48 h at 37 °C in a $CO_2$ incubator, assay was performed to examine the effect. For the luciferase reporter gene assay, the appropriate combination of pMIR-GTRAP3-18 3′-UTR (100 ng) or pMIR-NOVA1 3′-UTR plasmids (100 ng) with Renilla luciferase vector (pRL) (100 ng), siRNA (7.5 pmol), miRNA mimic (15 pmol), and miRNA inhibitor (45 pmol) was diluted in 25 μL of appropriated medium without serum in the well of the 24-well culture plate. Then diluted Lipofectamine RNAiMAX (1.5 μL) in 25 μL of the serum-free medium was added to each well and incubated for 10–20 min at room temperature. In the meantime, cells were diluted in growth medium containing the appropriate number of cells to give 30–50% confluence within 24 h after plating. Then 500 μL of diluted cells were added to each well. After cells were incubated approximately 48 h at 37 °C in a $CO_2$ incubator, assay was performed to examine the effect. For the transfection of Neuro2a cells, the following electroporation method with an NEPA 21 electroporator (Nepa Gene) was used. The cells were trypsinized, collected and washed with serum-free medium, and then placed in an electrode cuvette that was filled with 100 μL of serum-free medium containing an appropriate siRNA (30 pmol) or a combination of miRNA mimic (60 pmol) and inhibitor (180 pmol). The settings for the poring pulse were as follows: voltage, 125 V; pulse width, 2.5 ms; pulse interval, 50 ms; number of pulses, +2; and decay rate, 10%. The settings for the transfer pulse were as follows: voltage, 20 V; pulse width, 50 ms; pulse interval, 50 ms; number of pulses, ±5; and decay rate, 40%. Cells were harvested after washing with phosphate-buffered saline (PBS) twice at two days after transfection. According to the manufacturer's instructions, the guide strand of the miR-96-5p mimic is an unmodified RNA strand with a sequence corresponding exactly to the annotation in miRbase, which is 5′-UUUGGCACUAGCACAUUUUUGCU-3′. The guide strand of the negative control mimic, which has the sequence 5′-UCACCGGGUGUAAAU-CAGCUUG-3′, has no homology to any known miRNA or mRNA sequences in mice, rats, or humans. On the other hand, the miR-96-5p inhibitor is an antisense oligonucleotide with a sequence perfectly complementary to the mature miR-96-5p, which is 5′-GCAAAAATGTGCTAGTGCCAA-3′. The sequence of the negative control inhibitor, which is 5′-TAACACGTCTATACGCCCA-3′, has no hits of >70% homology to any sequence in any organism in the NCBI and miRBase databases.

**Western blotting**. Determination of the protein amount was performed using the BCA protein assay (Pierce), and the same amounts of proteins were normalized for total protein. The samples were boiled in RIPA buffer (20 mM Tris–HCl (pH 7.5), 150 mM NaCl, 1% NP-40, 1% sodium deoxycholate, 0.1% sodium dodecyl sulfate (SDS), and protease inhibitor cocktail (Sigma-Aldrich)), separated by SDS–polyacrylamide gel electrophoresis (PAGE), and transferred to polyvinylidene fluoride membranes (Bio-Rad). Non-specific binding was blocked with 3% skim milk in PBS-Tween20, and proteins were probed with anti-GTRAP3-18 (Novus Biologicals; NBP-84273) at 1:1000 dilution, anti-NOVA1 (Abcam; ab183024) at 1:1000 dilution, or anti-β-actin (Sigma-Aldrich; A5316) at 1:5000 dilution. After washing with PBS-Tween20, the horseradish peroxidase (HRP)-labeled secondary antibodies against rabbit (Chemicon; AP188P) or mouse IgG (Chemicon; AP181P) were probed and detected with an ECL prime HRP detection kit (GE Healthcare). Immunoblotting data was collected using a Luminograph I (ATTO) measuring emitted photons by chemiluminescence. Protein expression was then evaluated with CS analyzer4 (Ver. 2.2.3; ATTO).

**Immunocytochemistry**. We used the chloromethyl reagent 7-amino-4-chloromethylcoumarine (CMAC) (Life Technologies), which produces a highly fluorescent adduct upon reaction with thiol groups for the evaluation of intracellular GSH in SH-SY5Y cells. Cells were incubated at 37 °C for 15 min with 5 μM CMAC and then incubated with serum-free media for 30 min. The cells were fixed with 4% paraformaldehyde (PFA) and then permeabilized with 0.05% Triton-X100 in the case of multiple staining. Non-specific staining was blocked with the reagent PBS containing 5% BSA/0.1% Tween20, and the cells were then incubated with anti-GTRAP3-18 (Novus Biologicals; NB100-1105) and anti-NOVA1 (Abcam; ab183024) at 1:1000 dilution overnight at 4 °C. After a wash with PBS-Tween20, the cells were labeled with fluorescent-labeled secondary antibodies, Alexa-Fluor 488 anti-goat IgG (Molecular Probes; A11055), and Alexa-Fluor 647 anti-rabbit IgG (Molecular Probes; A31573) at 1:1000 dilutions. Finally, the cells were mounted using Fluoromount-Plus mounting solution (Diagnostic Biosystems) and captured with a Nikon A1 confocal microscope. The fluorescence intensity was analyzed by NIS-Elements Imaging Software (NIS Elements C (Ver. 4.30); Nikon) and an average intensity value of the cells in the field was calculated for further analysis.

**Luciferase reporter gene assay**. The 3′-UTR of GTRAP3-18 (NM_006407) containing a potential binding site for NOVA1 was amplified from cDNA of SH-SY5Y cells using forward primer 5′-GAGCTCACATAACTTACCTGAGCTAGG-3′ and reverse primer 5′-ACGCGTAAATAAAGTCTCACC-3′. The 3′-UTR of NOVA1 (NM_001366390) containing potential target sites for miR-96-5p was amplified from cDNA of SH-SY5Y cells using forward primer 5′-GAGCTCGTG CCCCAGTTACACATCAGA-3′ and reverse primer 5′-GCCGGCTGATGCTACA

TGATGAACTA-3′ (See also Supplemenatry Table 2). PCR products amplified with Prime STAR MAX (Takara Bio). The PCR condition was as follows: 30 cycles of 98 °C for 10 s, 55 °C for 15 s, and 72 °C for 10 s. Amplified products were cloned into the pMD20 T-vector using a Mighty TA-cloning kit (Takara Bio) and then confirmed by DNA sequencing (Eurofin). These inserts were then removed from the pMD20-T vector by SacI/MluI digestion (GTRAP3-18 3′-UTR) or SacI/NaeI digestion (NOVA1 3′-UTR), and subcloned into the firefly luciferase reporter vector, pMIR-REPORT (Promega). The mutation of the miRNA target sequence on GTRAP3-18 or NOVA1 3′-UTR was performed using a Mutagenesis kit (Takara Bio). The primer sequences for mutagenesis were as follows: GTRAP3-18 Ptbp1-mut, 5′-TCTTCTTGA CTTCTCAGACATGGTCTAGAATC-3′ (forward) and 5′-GACAAGTCAAGAAGATATCACTGTGTCTAAAGA-3′ (reverse); GTRA P3-18 unbound-mut, 5′-TGTCGTTGACTTGTCAGACATGGTCTAGAATC-3′ (forward) and 5′-GACAAGTCAACGACATATCACTGTGCTAAAGA-3′ (reverse); deletion-mut, 5′-CAGTGATAGACTTCACAGACATGGTC-3′ (forward) and 5′-GTGAAGTCTATCACTGTGCTAAAGATC-3′ (reverse); NOVA1 mut1: 5′-GAAAGTCGGAACAAATTATTGATAGCT-3′ (forward) and 5′-TTTG TTCCGACTTTCACTTTTGTTTAT-3′ (reverse); and NOVA1 mut2: 5′-CTGAG TCGGAACTGTCCAGGCCATTTG-3′ (forward) and 5′-ACAGTTCCGACTCAG GAGAGGTACAGA-3′ (reverse) (see also Supplemenatry Table 2). The PCR condition was as follows: 30 cycles of 98 °C for 10 s, 55 °C for 15 s, and 72 °C for 1 min 30 s. Cells were transfected with the appropriate combination of pMIR-GTRAP3-18 3′-UTR or pMIR-NOVA1 3′-UTR plasmids with Renilla luciferase vector (pRL), siRNA, miRNA mimic, and miRNA inhibitor using Lipofectamine RNAiMax (Life Technologies) as described in the Transfection method. Firefly luciferase activity was normalized to Renilla luciferase activity. Luciferase activity was measured by a Dual-luciferase Reporter Assay System (Promega) using a Luminometer TD-20/20 (Turner Biosystems).

**Intracellular ROS measurement**. The intracellular ROS in the Neuro2a cells was measured using an OxiSelect$^{TM}$ Intracellular ROS Assay Kit (Cell Biolabs). All experimental procedures were performed according to the manufacturer's instructions. Briefly, media was removed from all wells and discarded; cells were then gently washed twice with PBS. To each well, 100 µL of 2′,7′-dichlorofluorescin diacetate (DCFH-DA) solution diluted in serum-free medium was added, and then the plates were incubated at 37 °C for 1 h. The media was discarded and the cells were again washed twice with PBS. The cells were then treated with 0, 100, or 500 µM $H_2O_2$ for 20 min. Oxidation of DCFH by ROS and/or RNS results in a fluorescent derivative 2′,7′-dichlorofluorescein (DCF) whose fluorescence is proportional to the total ROS/RNS levels in the sample. Cells were lysed in 100 µL of cell lysis buffer and their fluorescence was monitored at 485 nm excitation/535 nm emission using a DTX 800 Multimode Detector (Beckman Coulter).

**Quantitative RT-PCR**. RNA isolation was carried out using Trizol Reagent (Life Technologies). We conducted reverse transcription (RT) on all individual RNA samples using a ReverTra Ace® qPCR RT Kit (Toyobo) with random hexamers as the RT primers, according to the manufacturer's protocol. Real-time polymerase chain reactions (PCRs) were performed using an Applied Biosystems 7500 Fast Real-Time PCR System (Thermo Fisher Scientific), and the amplifications were done using Fast SYBR® Green Master Mix (Thermo Fisher Scientific) with the following primer sets: for GTRAP3-18, 5′-GGAACAACCGTGTAGTGAGCAA-3′ (forward) and 5′-TGATGCCGAACACAAAGACC-3′ (reverse); for NOVA1, 5′-AGTATCCTACAACCTCAG-3′ (forward) and 5′-CTCCATTATAGCCTTCAC-3′ (reverse), and for GAPDH, 5′-AAAATGGTGAAGGTCGGTGTG-3′ (forward) and 5′-AATGAAGGGGTCGTTGATGG-3′ (reverse) (see also Supplementary Table 2). Cycling conditions were as follows; initial denaturation at 95 °C for 20 s, 40 cycles of denaturation at 95 °C for 1 s, annealing and extension at 60 °C for 20 s and followed by dissociation curve analysis.

**Animals**. Adult male C57Bl/6 mice (8 weeks old) were maintained under a 12/12 h light/dark (LD) cycle. To reduce the variation of data, all mice were starved for 1 day and then perfused intracardially with PBS under isoflurane inhalation anesthesia. All animal protocols were approved by the Animal Experimentation Committee of the Teikyo University School of Medicine.

**Protein identification**. The sequence of GTRAP3-18 3′-UTR was amplified from the luciferase reporter plasmid of GTRAP3-18 3′-UTR using forward primer 5′-TAATACGACTCACTATAGGGAGAACATAACTTACCTGA-3′ and reverse primer 5′-GGATCACTAGTAAGCTTAGATCTTAAATAAAGTCTCACC-3′ (see also Supplemenatry Table 2) with PrimeSTAR (Takara Bio). The PCR condition was as follows: 30 cycles of 98 °C for 10 s, 55 °C for 5 s, and 72 °C for 1 min 30 s. The PCR product was then digested with BglII and in vitro transcription was performed using CUGA7 RNA polymerase (Takara Bio). Transcribed RNA was desthiobiotinylated with an RNA 3′ end Desthiobiotinylation Kit (Pierce), and then conjugated with streptavidin-coupled Dynabeads (VERITAS). The brain extracts from mice were incubated with RNA conjugated magnetic Dynabeads at 4 °C overnight, and then the binding proteins were dissolved in the sample buffer (60 mM Tris-HCl (pH 6.8), 1% SDS, 10% glycerol, and 20 mM dithiothreitol) and loaded on SDS-PAGE gel. The gel was stained with Coomassie Brilliant Blue (CBB)

and the bands of candidate proteins were cut out and then subjected to LC-MS/MS analysis using Q Exactive Plus (Thermo Scientific) performed by Shimadzu Techno-Research.

**Intra-arterial injection**. One nmol of the miR-96-5p inhibitor or a negative control inhibitor (Exiqon) dissolved in 100 µL of artificial cerebrospinal fluid (aCSF) containing 130 mM NaCl, 3.5 mM KCl, 1.25 mM NaH$_2$PO$_4$, 2 mM MgSO$_4$, 2 mM CaCl$_2$, 20 mM NaHCO$_3$, and 10 mM glucose (pH 7.4) was mixed with 100 µL solution of MBs ($1 \times 10^9$ particles/mL) to make a 200 µL mixture. Mice were anesthetized by intraperitoneal injection of a mixture of 0.03 mg kg$^{-1}$ medetomidine, 5.0 mg kg$^{-1}$ butorphanol, and 4.0 mg kg$^{-1}$ midazolam. Under the stereoscopic microscope, a midline skin incision was made, and the subcutaneous tissue, fat, and muscles were dissected bluntly with forceps to expose the left common carotid artery, internal carotid artery, and external carotid artery. The common carotid artery was ligated with suture, a small incision was made on the common carotid artery by microscissors, and a microcatheter (Micro-Renathan tubing; Braintree Scientific) was inserted into and detained within the internal carotid artery. A 200 µL mixture of miRNA inhibitor and MBs was prepared and then administered with a microinjector (Eicom) at a programmed injection speed of 100 µL 2 min$^{-1}$. Simultaneously, a probe with a 12-mm diameter was placed on the shaved head of a mouse undergoing surgery and used to administer transcranial US (frequency: 3 MHz; intensity: 0.5 W cm$^{-2}$; exposure time: 2 min). After internal carotid artery infusion, the microcatheter was removed immediately. The insertion site was sealed with surgical glue, the ligature was removed, and common carotid artery reperfusion was visually confirmed. The ligation time of common carotid artery for the infusion was less than 5 min.

**Slice culture**. For the experiment on GSH detection in vivo, the brain was immediately cut into 300-µm-thick slices in gassed (95% oxygen, 5% CO$_2$) ice-cold aCSF after decapitation under isoflurane inhalation anesthesia. The experiments were initiated by transferring brain slices to tubes each containing aCSF containing 10 µM chloromethylfluorescein diacetate (CMFDA) at 30 °C for 30 min and transferred to fresh aCSF for an additional 30 min incubation at 30 °C under continuous bubbling with 95% oxygen/5% CO$_2$. Brain slices were incubated in aCSF.

Slices were fixed in 4% PFA and incubated overnight at 4 °C with a mouse anti-NeuN (Millipore; MAB377) at 1:1000 dilution. The slices were incubated with Alexa Fluor 546-conjugated goat anti-mouse IgG (Molecular Probes; A11003) at 1:1000 dilution. The sections were mounted using Fluoromount-Plus (Diagnostic Biosystems) and captured with a Nikon A1 confocal microscope.

**Preparation of lipid-based microbubbles**. 1,2-distearoyl-sn-glycero-3-phos-phocholine (DSPC), 1,2-distearoyl-sn-glycero-3-phosphoglycerol (DSPG) and N-(carbonyl-methoxypolyethyleneglycol 2000)-1,2-distearoyl-sn-glycero-3-pho-phoethanolamine (DSPE-PEG2000) were purchased from the NOF Corporation. Perfluoropropane (C$_3$F$_8$) was purchased from Takachiho Chemical Industrial Co.

Liposomes composed of DSPC, DSPG, and DSPE-PEG2000 in a molar ratio of 30:60:10 were prepared by the lipid film hydration method[51,52]. In brief, all lipids were dissolved in chloroform, methanol, ammonium solution, and water (65:35:4:4, volume ratio, respectively). The lipid film was prepared with a rotary evaporator, following by drying overnight in a vacuum desiccator to completely remove the solvents. The lipid film was then hydrated with 100 mM phosphate buffer (pH 7.4) at 65 °C with shaking, and then liposomes were sonicated with a bath-type sonicator (Bransonic 2150j-DTH; Branson Ultrasound Co.). For MB formation, the liposome suspension (lipid concentration: 1 mM, 300 mL) was homogenized at 7500 rpm for 60 min at 40 °C under a C$_3$F$_8$ atmosphere with a homogenizer (Labolution Mark II 2.5; Primix Corporation); the mixing head was a rotor stator-type mixer with a 30 mm stator inside-diameter (Awaji). The MB suspension was mixed with 18% sucrose solution with a 1: 1 volume ratio, and 2 mL of mixture was dispensed in a 5 mL vial. The air in the head space was replaced with C$_3$F$_8$, and the vial was closed with a rubber lid, followed by freezing at −30 °C. After freezing and opening the rubber lid, freeze-drying was performed at −30 °C for 1 h, −20 °C for 72 h, and 20 °C for 48 h using a shelf temperature controlled drying chamber (EYELA FDU-1100 and DRC-1100; EYELA). After the drying was completed, the sample chamber was filled with C$_3$F$_8$ and the vials were closed with a rubber lid, after which the vials were taken out and sealed with aluminum caps. The vials were kept at room temperature until each experiment. Before the experiments, the freeze-dried MBs were reconstituted with 2 mL of water and briefly shaken. Mini-Spike filters with a 5 µm cut-off (B. Braun AG; Melsungen) were used when withdrawing the MB suspension to remove any large MBs. The number of MBs was measured with a Coulter Counter (Multisizer 3; Beckman Coulter). Finally, the concentration of MBs was adjusted to $1 \times 10^9$ particles mL$^{-1}$ with PBS, and the MB suspension was utilized in each experiment.

**Immunohistochemistry**. Mice were perfused with PBS containing 4% PFA after saline perfusion to remove blood under isoflurane inhalation anesthesia. Their brains were then placed in an optimal cutting temperature compound and frozen with liquid nitrogen. Coronal brain sections were cut on a cryostat at 20 µm thickness and stored at −80 °C. Antigen retrieval using Immunosaver reagent

(Nisshin EM) was performed for 45 min with heating at 98 °C in an electric kitchen pot before starting immunohistochemistry. The slices were placed in blocking reagent (PBS containing 1% BSA/0.2% TritonX-100/1 mM EDTA) after permeabilizing with 0.05% TritonX-100 treatment and then incubated overnight at 4 °C with anti-EAAC1 (Abcam; ab124802) at 1:200 dilution, anti-GTRAP3-18 (Novus Biologicals; NB100-1105) at 1:200 dilution, anti-NOVA1 (Abcam; ab183024) at 1:1000 dilution, anti-Iba1 (Wako; 019-19741) at 1:1000 dilution, anti-GFAP (Sigma; G-3893) at 1:1000 dilution, anti-LC3 (MBL; PM036MS) at 1:1000 dilution or anti-4-HNE (abcam; ab48506) at 1:1000 dilution. After a wash with PBS-Tween20, the slices were labeled with appropriate fluorescent-labeled secondary antibodies, Alexa-Fluor 488 anti-rabbit IgG (Molecular Probes; A11008), Alexa-Fluor 488 anti-goat IgG (Molecular Probes; A11055), Alexa-Fluor 594 anti-mouse IgG (Molecular Probes; A11005) or Alexa-Fluor 647 anti-rabbit IgG (Molecular Probes; A31573) at 1:1000 dilutions. For nuclear labeling, 4′,6-diamidino-2-phenylindole (DAPI) (Dojindo) was used. The section was mounted using Fluoromount-Plus (Diagnostic Biosystems) and captured with a Nikon A1 confocal microscope.

**Statistics and reproducibility**. Data were shown as mean value ± standard deviation (SD) obtained from biologically independent samples. All experiments were repeated at least three independent experiments and precise sample number was indicated in the "Result" section and figure legends. Data were analyzed by one-way ANOVA for multiple comparison and the appropriate statistical test noted in the figure legends was used for the analysis of siRNA, miRNA mimic, or miRNA inhibitor effects. Significance in difference between two groups were tested by Student's $t$-test and Tukey's HSD test was performed for a comparison of more than two groups. IBM SPSS Statistics (Rel. 24.0.0.0) was used for statistical analysis. Differences with $p < 0.05$ were considered significant. Estimates of effect sizes was measured by calculating Cohen's $d$ and 95% confidence intervals (CI) for $d$ were also shown. Effect size was considered large when $|d| \geq 0.8$.

**Reporting summary**. Further information on research design is available in the Nature Research Reporting Summary linked to this article.

## Data availability

All source data for the graphs and tables are listed in Supplementary Data 1–7. Supplementary Data 1 contains the source data underlying Fig. 1. Supplementary Data 2 contains the source data underlying Fig. 3. Supplementary Data 3 contains the source data underlying Fig. 4. Supplementary Data 4 contains the source data underlying Fig. 5. Supplementary Data 5 contains the source data underlying Supplementary Table 1. Supplementary Data 6 contains the source data underlying Supplementary Table 2. Supplementary Data 7 contains the source data underlying Supplementary Figs. 2, 4, 5, 6, and 7. The other datasets generated/analyzed during the current study are available from the corresponding author (T. N.) on reasonable request.

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

## Acknowledgements

We are grateful to Dr. H. Tanaka and Dr. A. Tamura for technical support. We also thank Dr. W. Sumida and Dr. M. Sugii for technical assistance and Y. Arai for secretarial assistance. This work was supported by JSPS KAKENHI Grant Numbers 15K18996 and 17K18122, AMED under Grant Number 17dm0107115h002, the Kowa Life Science Foundation, the Naito Science & Engineering Foundation and the Research Fund of Teikyo University.

## Author contributions

C.K. and K.A. designed the experiments and wrote the manuscript. T.N. supervised the whole project. K.K.U. performed i.a. injection. D.O., K.M., and R.S. prepared lipid-based microbubbles and provided technical information for ultrasound application. C.K. performed all other experiments and statistical analysis. N.M. and Y.O. provided experimental support for the in vivo and in vitro experiments.

## Competing interests

The authors declare no competing interests.
