## [Peer Review File · Communications Biology]

Reviewers' comments:

Reviewer #1 Attachment 
Reviewer #2 (Remarks to the Author):

The authors have followed up on their previous work showing that glutathione levels in different cell types are regulated by miR-96-5p through its dual action on EAAC1 (previous work) and GTRAP3-18 via the intermediary NOVA1 protein. They have further corroborated this in vivo, by delivering the miR-96-5p inhibitor in mice through the use of ultrasound-mediated BBB opening and shown an increase in GSH levels. The results are presented in a comprehensive way and are of particular interest for further studies related to oxidative stress in various neurodegenerative disorders as well as novel therapeutic targets. 
Reviewer #3 (Remarks to the Author):

In this study, the authors investigate the function of miR-96-5p inhibitor in increasing the GSH in brain. Overall, the study is sound. There are few issues that the authors may consider before publication.

1. In the abstract, it is mentioned as "The microRNA miR-96-5p increases EAAC1 and decreases GTRAP3-18." However, it is still written as, I quote, "intra-arterial injection of an miR-96-5p-inhibiting nucleic acid to living mice ... decreased the level of GTRAP3- 18 via NOVA1 and increased the levels of EAAC1 and GSH ..." Should these two statements are contradictory?
2. A computational analysis was performed to confirm that the 3'-UTR sequence of GTRAP3-18 has target sites of miR-96-5p. Can this also be tested by experimental means?
3. What are the criteria for selecting the five bands?
4. For figure 2 and so on, would it worth to have another group where only the inhibitor is added? As this seems a more direct way to see if the inhibitor functions in the expected way.
5. The last sentence of the Section "Preparation of lipid-based microbubbles" of Methods, 'B' is supposed to be "MB"?
6. Figure 4 is cited before Figure 2, and 3. Would the order of figures can be adjusted in some way?
7. There is no page number or line number, marking it is a bit hard for writing this review report.

Reviewer #1

We are grateful to reviewer 1 for the insightful comments and useful suggestions that have helped us to improve our paper. As indicated in the responses that follow, we have taken all these comments and suggestions into account in the revised version of our paper.

Comment 1. Abstract: please reformulate the statement in the lines 2-4

Response: As you suggested, we reformulate the statement as “Glutathione (GSH) is an important antioxidant that plays a critical role in neuroprotection. GSH depletion in neurons induces oxidative stress and thereby promotes neuronal damage, which in turn is regarded as a hallmark of the early stage of neurodegenerative diseases.” in page 2 line 27-30.

Comment 2. Introduction: some statements are generic and require the insertion of references; and (also) I suggest to re-read accurately to avoid repetitions.

Response: As you suggested, we deleted the sentence “Postmortem studies have shown that brain GSH levels are significantly reduced in patients with NDs.” in page 3 line 60. We also changed the sentence in page 5 line 101-102 to “So far, NOVA1 has been reported to be involved in mRNA processing, splicing and miRNA regulation.” deleting the word “RBP” to avoid the repetition.

In addition, we added several references as follows.

- “Heneka, M. T., Kummer, M. P. & Latz, E. Innate immune activation in neurodegenerative disease. *Nature reviews. Immunology* 14, 463-477, doi:10.1038/nri3705 (2014).” in page 3 line 53.
- “Mischley, L. K. et al. Glutathione as a Biomarker in Parkinson's Disease: Associations with Aging and Disease Severity. *Oxidative medicine and cellular longevity* 2016, 9409363, doi:10.1155/2016/9409363 (2016). “ and “Mandal, P. K., Saharan, S., Tripathi, M. & Murari, G. Brain glutathione levels--a novel biomarker for mild cognitive impairment and Alzheimer's disease. *Biological psychiatry* 78, 702-710, doi:10.1016/j.biopsych.2015.04.005 (2015).” in page 3 line 64.
- “Wong, Y. C. & Krainc, D. α -synuclein toxicity in neurodegeneration: mechanism and therapeutic strategies. *Nature medicine* 23, 1-13, doi:10.1038/nm.4269 (2017).” in page 4 line 91.
- “Khong, A. & Parker, R. The landscape of eukaryotic mRNPs. *RNA (New York, N.Y.)* 26, 229-239, doi:10.1261/rna.073601.119 (2020).” in page 5 line 96.

- “Banks, W. A. From blood-brain barrier to blood-brain interface: new opportunities for CNS drug delivery. *Nature reviews. Drug discovery* 15, 275-292, doi:10.1038/nrd.2015.21 (2016).” in page 5 line 107.
- Daneman, R. & Prat, A. The blood-brain barrier. *Cold Spring Harbor perspectives in biology* 7, a020412, doi:10.1101/cshperspect.a020412 (2015). in page 5 line 109.

Comment 3-#1. Experiments showed in figure 1 relative to the investigation of indirect regulation of GTRAP3-18 by miR-96-5p are performed in HEK cells. I strongly recommend the confirmatory experiments in a neural cell line (SHSY5Y? SK cells or maybe in differentiated iPS).

Response: In accord with this suggestion, we performed same experiments using mouse neuroblastoma cell line Neuro2a and obtained similar results with the ones using HEK293 cells. HEK293 results were moved to Supplementary Figure S3 and Neuro2a results were added in the Figure 1c and d, instead.

Comment 3-#2. Exp reported in Figure 1b: if the hyp is that the miRNA is on the UTR of the GTRAP3-18 RNA the RNA must be quantified by real time.

Response: In accord with this suggestion, we performed quantified RT-PCR using transfected Neuro2a. The results were added in the Supplementary Figure S4.

Comment 3-#3. Figure 4a is showed before Figure 1c and other figures. Please reorganize the results presentation. This is quite confusing for the reader

Response: In accord with this suggestion, we reorganized the results and figures.

Comment3-#4. The bioinformatic analysis must be anticipated. I think that this is prodromal for wet experiments and results

Response: We used bioinformatic analysis in this manuscript for the prediction of miR-96-5p target site on 3'-UTRs of GTRAP3-18 and NOVA1 3'-UTR, and NOVA1 binding site on GTRAP3-18 3'-UTR. These bioinformatic analysis were confirmed by luciferase reporter gene assay using mutant luciferase constructs of their 3'-UTR.

In accord with suggestion, we added the sentence as “which was consistent with the

bioinformatical prediction” or “which was not consistent with the bioinformatical prediction” in page 12 line 246-247, page 12 line 250 and page 13 line 285.

Comment3-#5. Again, the investigation of regulation of NOVA1 expression by miR-96-5p id performed in bHEK93 cells...as before please confirm these findings in neural context.

Response: In accord with this suggestion, we performed same experiments using mouse neuroblastoma cell line Neuro2a and obtained similar results with the ones using HEK293 cells. HEK293 results were moved to Supplementary Figure S5 and Neuro2a results were added in the Figure 4a, b and c, instead.

Comment3-#6. I suggest to organize the experiments presented in figure 3....panel 3c comes before of panel 3b!!! Why??? And, moreover, respect to the results presented in panel 3c I strongly suggest a more quantitative approach not only fluorescence quantification...

Response: In accord with this suggestion, we reorganized the results and figures. In addition, we basically quantified the protein expression in the cell culture experiments by western blotting analysis. The results of protein expression from fluorescence experiment was moved to Supplementary Figure S1 and S5d.

Comment3-#7. The in vivo study is extremely interesting BUT why they studied only the dentate gyrus??? It would be interesting to see the effects of miR-96-5p inhibitor also in other brain areas...(striatum?? SNpc; cortex??)

Response: Because the diameter of ultrasound probe is 12-mm, the area of ultrasound irradiation is limited around dentate gyrus of hippocampus. The effects of miR-96-5p inhibitor was not always observed in other areas of brain.

As you suggested, we described as follows “The effect of the miR-96-5p inhibitor was particularly prominent in the DG of the hippocampus, possibly because of the limited US exposure area.” in Discussion section; page 20 line 412-413.

Comment 4. Discussion: require some improvements in terms of references citations.

Response: In accord with this suggestion, we added following references;

- “Lewis, H. A. et al. Crystal structures of Nova-1 and Nova-2 K-homology RNA-binding domains. *Structure* (London, England : 1993) 7, 191-203, doi:10.1016/s0969-2126(99)80025-2 (1999).” in page 18 line 382
- “Kedde, M. et al. A Pumilio-induced RNA structure switch in p27-3' UTR controls miR-221 and miR-222 accessibility. *Nature cell biology* 12, 1014-1020, doi:10.1038/ncb2105 (2010).” in page 19 line 395 and 397.
- “Jordão, J. F. et al. Antibodies targeted to the brain with image-guided focused ultrasound reduces amyloid-beta plaque load in the TgCRND8 mouse model of Alzheimer's disease. *PLoS One* 5, e10549, doi:10.1371/journal.pone.0010549 (2010).” in page 20 line 421.
- “Nisbet, R. M. et al. Combined effects of scanning ultrasound and a tau-specific single chain antibody in a tau transgenic mouse model. *Brain : a journal of neurology* 140, 1220-1230, doi:10.1093/brain/awx052 (2017).” in page 20 line 421.

Comment 5. I also have some concerns on methods: why the authors decided to starve animals?

Response: We used starved animals because our unpublished data was shown that the variation in the raw data is less than ad lib animals possibly because of nutrient absorption. In accord with this suggestion we added following sentence “To reduce the variation of data, all mice were starved for 1 day...” in the Methods section in page 27 line 560.

Reviewer #2

We would like to express our appreciation to reviewer 2 for the constructive suggestions. As indicated in the responses that follow, we have taken all these comments and suggestions into account in the revised version of our paper.

Comment 1. The presentation of figures can be drastically improved. It does not help that the reader has to jump between figures while reading the text, as well as only look at a part of a figure. This can get very confusing at times. The authors should try and revise their figures.

Response: In accord with this suggestion, we reorganized the results and figures.

Comment 2. There is some data missing: page 10, line 10 “this reduction was again blocked by the miR-96-5p inhibitor (P<0.05 versus the effect of inhibitor by Tukey’s HSD test).” This experimental dataset is not shown (pertaining to fig 4e)

Response: In accord with this suggestion, we added figure number after the sentence in page 11 line 239.

Comment 3. For Fig 2c, the differences in relative CMAC level by IF are not clear in the figure. As such, the authors can either change the representative image to better reflect the changes, or add a zoomed panel to show the cells at higher magnification.

Response: In accord with this suggestion, we changed the image with higher magnification as shown in Supplementary Fig. S1. and also added it with enlarged panels in Fig 1c of which number have been changed from Fig.2c in accord with the reviewers’ suggestion.

Comment 4. For all the imaging figures, the author do not describe in their methods section how the fluorescence was analysed (whole image intensity or per cell analysis). This might help understand why there is a discrepancy between the analysis and the figure.

Response: In accord with this suggestion, we added following sentence in the Methods section “The fluorescence intensity was analyzed by NIS-Elements Imaging Software

(Nikon) and an average intensity value of the cells in the field was calculated for further analysis.” in page 24 line 500-502.

Comment 5. The authors have mentioned that transfection with miR-96-5p increase oxidative stress, however they have not measured this. It would be interesting to see this effect and its reduction with the miRNA inhibitor.

Response: In accord with this suggestion, we performed additional experiment which detect intracellular reactive oxygen species (ROS). As a result, ROS was significantly increased with the transfection of miR-96-5p mimic in the 500uM H₂O₂-treated Neuro2a cells. In addition, miR-96-5p inhibitor could decrease the ROS level in the 100uM H₂O₂-treated Neuro2a cells. The results were added in the Supplementary figure S2 and also mentioned in page 8 line 149-160.

Comment 6. The authors state that the YCAY clusters is important for the stability of GTRAP3-18

mRNA or protein. Considering this, mutations in this regions (as the authors have performed), might not be indicative of the actual effect of NOVA1 binding and regulation as they have seen similar effects when this cluster was deleted. Have the authors thought of any rescue experiments?

Response: We hypothesized that the binding of some proteins is important for GTRAP3-18 mRNA stability. We made a new mutation construct which can be bound with the other RNA binding protein Ptbp1 but not NOVA1. As a result, the mutated GTRAP3-18 is more stable than non-mutated one. In addition, the effect of miR-96-5p is completely vanished. This result was added in the Fig.4d and e.

Comment 7. Ultrasound as a technique can itself lead to oxidative stress (often seen with astrogliosis and microglial activation; ref Jalali 2010, BMC neurology and Kovacs 2017,PNAS). It will be interesting to see if there is a reduction in oxidative stress caused by BBB opening, with the administration of the miRNA inhibitor.

Response: In accord with this suggestion, we performed additional immunohistochemical experiment using the antibodies against Iba1 as a marker for microglial activation and GFAP as a marker for glial activation. As a result, Iba1 expression was increased with Ultrasound application compared to sham-operation.

Furthermore, the expression of Iba1 in the brain of mice with the administration of miR-96-5p inhibitor is less than the negative control. These data are shown in Supplementary figure S9 and mentioned in page 16 line 346-354.

In addition, we detected 4-HNE as an indicator of lipid peroxidation caused by oxidative stress. However, there is no difference between miR-96-5p inhibitor and negative control. It might be necessary to treat with oxidant agents for the detection of 4-HNE. This result is added in Supplementary figure S10 and mentioned in page 16 line 354-356.

Comment 8. What are the other effects of miR-96-5p? It is known that this miRNA is associated with reduced apoptosis and autophagy in many cancers (through PTEN and mTOR-AKT pathways). Have the authors looked at autophagy or apoptosis markers following the inhibition of miR-96-5p in the brain? This is especially important in neurodegenerative disorders which have defects in clearing protein aggregates. Ultrasound has also been shown to induce autophagy and clear tau aggregates (Pandit et al, 2019, Theranostics), so it will be interesting to see if the inhibitor has a synergistic effect of inducing autophagy with ultrasound.

Response: In accord with this suggestion, we performed additional immunohistochemical experiment using the antibodies against LC3 as a marker for autophagy. As a result, LC3 expression in the brain of mice with the administration of miR-96-5p inhibitor is increased compared with negative control and sham operation. This data was added in Supplementary figure S10 and mentioned in page 16 line 356-360.

The manuscript could be improved by certain minor revisions:

Comment 1. The introduction has a few repetitions: for example sentences 12 and 15. The introduction could also benefit with a much clearer explanation of the role of GSH in the brain as a neuroprotective agent.

Response: In accord with this suggestion, we deleted sentence 12 and reformulate sentence in Introduction section. In addition, we added the explanation of the role of GSH in the brain as a neuroprotective agent by adding following sentence “GSH acts as an electron donor and thereby detoxifies either ROS or RNS by oxidizing itself.” in page 3 line 59-60.

Comment 2. The authors have mislabelled figures 2c and d as fig 4c and d (page 11, line 5, pertaining to the CMAC figure).

Response: We appreciate that you pointed out our mistake. In accord with this suggestion, we precisely checked figure number and corrected in a right way.

Comment 3. The authors should mention that supplementary figure 2 depicts the analysis for figure 3d (page 11, line 21)

Response: In accord with this suggestion, we reformulated the legend of Supplementary Figures.

Comment 4. The authors have injected the inhibitor drug via intra-arterial injection. This mode of delivery has been shown to have some limited success for brain delivery of small drugs. This is also observed in this study to some extent when comparing the CMFDA signal in the nontreated hemisphere of the mice injected with the inhibitor. The authors argue that improvement and modification of microbubble technologies will be needed before IV injections can be used in place of i.a. administration from the ICA. While the carotid artery supplies to the brain, the process of intra-arterial injection requires surgery and in general is tedious. The authors might want to try IV or retro orbital delivery, as these modes have been used previously for drug delivery and shown to be safe (Jordao et al, 2010, PLoS One for IV administration of anti-amyloid antibodies; Nisbet et al, 2017, Brain for retro-orbital administration of scFv against phosphor-tau)

Response: In accord with this suggestion, we reformulated the Discussion by adding following sentence “We thus succeeded in delivering the miR-96-5p inhibitor under somewhat limited conditions, but further research is needed to solve the problem of delivering a miR-96-5p inhibitor to a limited brain area. In addition, i.a. administration requires advanced surgery, which is generally inappropriate for clinical treatment. Improvement and modification of MB techniques will be needed before intravenous or retro-orbital injection can be used in place of i.a. administration from the ICA.” in page 20 line 416-421 and added recommended references.

Reviewer #3

We would like to thank reviewer 3 for the thoughtful comments. As indicated in the responses that follow, we have taken all these comments and suggestions into account in the revised version of our paper.

Comment 1. In the abstract, it is mentioned as “The microRNA miR-96-5p increases EAAC1 and decreases GTRAP3-18.” However, it is still written as, I quote, “intra-arterial injection of an miR-96-5p-inhibiting nucleic acid to living mice ... decreased the level of GTRAP3- 18 via NOVA1 and increased the levels of EAAC1 and GSH ...” Should these two statements are contradictory?

Response: You thankfully pointed out our mistake. We made a correction and reformulate the statements as “In this study, we found that the GTRAP3-18 level was increased by up-regulation of the microRNA miR-96-5p, which was found to decrease EAAC1 levels in our previous study.”, shown in page 2 line 32-34.

Comment 2. A computational analysis was performed to confirm that the 3'-UTR sequence of GTRAP3-18 has target sites of miR-96-5p. Can this also be tested by experimental means?

Response: This can be tested by making mutation luciferase constructs of GTRAP3-18 3'-UTR, which cannot be bound with mediator protein of miR-96-5p, NOVA1; and performing luciferase reporter gene assay. As a result, luciferase activity is not changed between miR-96-5p mimic transfection and negative control when NOVA1-unbound luciferase constructs were used. These results indicate that miR-96-5p does not have the effect of GTRAP3-18 transactivation and/or stability without NOVA1 binding suggesting no target site of miR-96-5p on GTRAP3-18 3'-UTR. These results are shown in the Supplementary Fig. S6 and S7.

Comment 3. What are the criteria for selecting the five bands?

Response: We added criteria for selecting the five bands and added description as “Among them, five bands which showed a more than 2-fold increase in the lane for GTRAP3-18 binding compared to the control were selected as candidate binding proteins for further analysis.” in page 10 line 210-212.

Comment 4. For figure 2 and so on, would it worth to have another group where only the inhibitor is added? As this seems a more direct way to see if the inhibitor functions in the expected way.

Response: In accord with this suggestion, we added the inhibitor only result in corresponding figure.

Comment 5. The last sentence of the Section “Preparation of lipid-based microbubbles” of Methods, ‘B’ is supposed to be “MB”?

Response: We appreciate that you pointed out our mistake. We corrected the word “B” to “MBs”.

Comment 6. Figure 4 is cited before Figure 2, and 3. Would the order of figures can be adjusted in some way?

Response: In accord with this suggestion, we reorganized the results and figures.

Comment 7. There is no page number or line number, marking it is a bit hard for writing this review report.

Response: In accord with this suggestion, we added page number and line number.

REVIEWERS' COMMENTS:

Reviewer #1 (Remarks to the Author):

The authors have performed a good job. In this new form the paper is sound. Please accept the manuscript for publication

Reviewer #2 (Remarks to the Author):

The authors have successfully addressed almost all the concerns raised in the initial review. The revised manuscript is a well-written and commendable work on the regulation of GSH via NOVA1 and an miRNA inhibitor. Furthermore they have also shown its in vivo application with focussed ultrasound as a delivery agent. The results are interesting for neurobiology and therapeutics.

This article is suitable for publication in its current form.

Reviewer #1

We are grateful to reviewer #1 for the critical comments and useful suggestions that have helped us to improve our manuscript.

Comment: The authors have performed a good job. In this new form the paper is sound.

Please accept the manuscript for publication

Reviewer #2

We appreciate reviewer #2 for identifying areas of our manuscript that needed corrections or modification.

Comment: The authors have successfully addressed almost all the concerns raised in the initial review. The revised manuscript is a well-written and commendable work on the regulation of GSH via NOVA1 and an miRNA inhibitor. Furthermore they have also shown its in vivo application with focussed ultrasound as a delivery agent. The results are interesting for neurobiology and therapeutics.

This article is suitable for publication in its current form.